# Review of Environmental and Health Factors Impacting Captive Common Marmoset Welfare in the Biomedical Research Setting

**DOI:** 10.3390/vetsci10090568

**Published:** 2023-09-12

**Authors:** Monika Burns

**Affiliations:** Novartis Institutes for BioMedical Research, Cambridge, MA 02139, USA; monika.burns@novartis.com

**Keywords:** nonhuman primate, NHP, marmoset, welfare, veterinary, husbandry, animal model

## Abstract

**Simple Summary:**

Common marmosets (*Callithrix jacchus*) are a small neotropical species of nonhuman primate that are commonly kept in captivity in zoological and biomedical research settings. The use of nonhuman primates in biomedical research has been critical to many advances that have allowed for the treatment of human diseases. The use of marmosets in research has increased in recent years. While there are some publications that describe best practices for their care and use in biomedical research settings, there is a need to develop additional reference guidelines. This manuscript reviews and summarizes publications related to the unique needs of marmosets kept in biomedical research settings and provides a comprehensive review of factors that should be considered by all staff working with these animals in order to promote optimal health and research use. This manuscript also highlights environmental and other factors that may impact marmoset welfare and provides recommendations on how best to plan for, care for, and work with marmosets in a biomedical research setting.

**Abstract:**

As a small-bodied neotropical nonhuman primate species, common marmosets have unique requirements for adequate husbandry and veterinary care to ensure proper maintenance and to promote good animal welfare in a biomedical research setting. Environmental conditions, as well as medical and research-related manipulations, can impact marmoset welfare. Research focus areas, including basic neuroscience, transgenics, and aging, involve additional implications for marmoset welfare. This manuscript provides a comprehensive review of factors that should be considered and mitigated as needed by clinical and research staff working with marmosets in biomedical research facilities to optimize the welfare of captive marmosets.

## 1. Introduction

The use of marmosets in biomedical research and the maintenance of the species in research animal vivaria has increased in recent years due to a number of factors that contribute to the marmoset being an attractive research model. As usage grows, it is critical for all members of the marmoset user community to consider factors of housing and husbandry, veterinary care, and research use that may impact animal welfare during preparation and continued support of marmoset colonies. This publication will review these critical factors and propose areas of future research that will be required to further promote the welfare of marmosets maintained in biomedical research facility settings. It will also provide practical guidance and recommendations for best practices for marmoset users to incorporate into their programs to optimize animal welfare. 

Nonhuman primates of many species have long been used in basic and preclinical biomedical research due to their high rates of homology to humans. While macaque species, including rhesus and cynomologus monkeys, have historically been the most heavily used in biomedical research, usage of the common marmoset (*Callithrix jacchus*) has increased globally in recent years. This is due at least in part to the fact that marmosets are viewed as having certain distinct advantages over the macaque nonhuman primate model. These perceived advantages include a small body size compared to macaque species, high reproductive fecundity, a relative lack of occupational health issues due to their inability to subclinically carry *Macacine herpesvirus* 1, and relatively accelerated lifespan/aging cycle compared to macaque species [1]. Historically, marmosets have been used in wide-ranging fields of research including infectious disease, neuroscience, aging and obesity, transgenics, and toxicology, among others [2,3]. Many aspects of marmoset biology and anatomy are unique when compared to macaques and must be accommodated in the research setting in order to assure proper use and promote animal welfare. 

It is critical for all marmoset users to promote the psychological well-being and welfare of all animals housed in their facilities. Making welfare a priority mitigates ethical concerns about the use of animals in research to the fullest extent possible and also ensures that scientific data produced during the study phase will be as accurate and translatable as possible as good animal welfare is key to good data [4,5]. Marmoset users should aim to optimize welfare by reducing stress and providing opportunities to engage in species-specific behaviors. While some studies focused on the welfare of Old World primate (OWP) species commonly engaged in biomedical research exist in the literature, primarily rhesus and cynomolgus macaques, the literature is lacking for marmosets [4,6]. Marmosets are susceptible to different stressors in captivity than OWPs and have unique species-typical behaviors that they engage in to reduce stress, which must be accommodated by housing and handling practices. They also are susceptible to common medical conditions that may impact health and welfare that do not routinely occur in OWP colonies [7]. Therefore, while some welfare standards and practices employed to maintain OWP colony welfare can be translated to marmosets, there are unique needs that must be addressed and mitigated to best maintain marmosets. Recommendations and suggested refinements that can aid marmoset users in improving animal health and welfare are outlined throughout the manuscript.

## 2. Facility, Housing, and Husbandry

### 2.1. Sourcing of Animals, Transport, Housing

Creating conditions that promote good marmoset welfare begins well before animal arrival. When establishing or adding animals to a colony, it is critical to source new animals from vendors that follow ethical practices of animal care. Ideally, the acquiring institution should conduct an animal welfare, animal health, and biosecurity-focused assessment of the vendor that provides insight into all care and use practices. This review of facilities and standard practices will allow the acquiring institution to make an assessment as to whether the vendor meets the criteria for purchase. Medical records should be thorough, complete, and readily shared with the acquiring facility. Tools such as the Animal Care & Use Program Assessment Tool for Contract Research Organizations published by the International Consortium for Innovation & Quality in Pharmaceutical Development can be used or adapted for this purpose [8]. For international imports, animals should be maintained in a quarantine facility that has experience in marmoset husbandry and care, and conditions in the quarantine and at the receipt facility should mimic those at the facility of origin during the transition period to allow the animals to slowly acclimate to their new environment. Diet offerings should be kept as consistent as possible during periods of high stress, such as shipment and quarantine, and ideally few changes should be made to the diet in the immediate period post shipment. Since the specific vendor facility affects the composition of the marmoset gut microbiome, it is possible that animals originating from different facilities may respond differently to the same new diet [9]. Facilities should expect to prepare extensively for the arrival of a new marmoset colony, especially if the facility has never housed marmosets before. Suggested actions that marmoset users should consider prior to animal arrival are outlined in Table 1.

### 2.2. Animal Transport and Quarantine

It is important that transporters experienced with unique species requirements be used for marmoset transport. It is also critical that only animals in good health condition be shipped. The acquiring institution can use the facility assessment process to feel confident that final clearance of animals for shipment is conducted appropriately during pre-shipment medical exams. Animals must be shipped in accordance with local, national, and international regulations related to live animal transport. When separated for shipment, animals should be socially housed as soon as possible with compatible partners at the destination facility. Infants should remain with dams during all transport and long transport events are not recommended for dam–infant pairs. Recommendations for quarantine practices are outlined in Chapter 10 of *The Common Marmoset in Captivity and Biomedical Research* [10]. The author recommends use of a formal monitoring system during the initial weeks of quarantine with a minimum of two daily recorded observations to ensure timely interventions, including clinical support, are made as needed. General guidelines for continued health monitoring of NHP colonies and specifically marmoset colonies have been published previously [10,11]. Adequate transport, quarantine, and health monitoring practices are key to maintaining healthy marmoset colonies, which supports overall animal well-being.

### 2.3. Facility and Housing Environment

In their natural environments, common marmosets live in many diverse habitats that can be difficult to replicate exactly in captivity. Reducing stress related to the physical cage environment and husbandry practices is essential to promoting welfare within a captive marmoset colony. While it may be tempting, especially for new marmoset users, to adapt previously existing dog or macaque housing for marmoset use, the author does not recommend this practice as the housing needs of marmosets are unique. There is great variety in the floor space, height, and volume requirements for marmosets depending on the animal welfare legislation of each country or region. *The Common Marmoset in Captivity and Biomedical Research* describes these regulatory requirements as well as commonly used internal cage space volume measurements in Tables 5.1 and 5.2, respectively [12]. Further information about caging at National Primate Research Centers in the United States can be found in Layne et al. [13]. Marmoset users should purchase and use either marmoset caging from established animal caging vendors or custom-design caging engineered to be appropriate for marmoset needs from a reliable manufacturer.

Social housing as a default housing condition is as critical for marmosets as it is for all primate species. If animals must be individually housed for experimental reasons, accommodations should be made to their cage environment to promote a good welfare state. Use of an enhanced environmental enrichment program for singly housed marmosets may decrease cortisol levels [14]. Other environmental factors, such as the number of animals in a cage, the specific social housing arrangement, the total number of animals and cages in a room, the presence of environmental noise, and proximity to other species, may impact overall stress levels as well. As a neotropical species, marmosets are housed both in zoological and research settings in outdoor or indoor/outdoor settings in warmer parts of the world or during warm months of the year in cooler climates. Where it is possible to house marmosets in large outdoor enclosures, a recent study has shown that there is no additional veterinary risk to doing so [15]. Common marmosets in the wild may adapt to living in cooler temperatures, However, the species is typically housed at temperatures between 75 and 81F, though this may be below the species’ thermoneutral zone [12]. Access to social, physical, and food enrichment were identified as a top indicators of marmoset welfare in a recent survey of marmoset users. While a limitation of this survey was the low number of respondents (*n* = 6), the high degree of agreement between respondents suggests that these enrichment opportunities should be provided to all marmosets [4].

Changing cage configuration can result in a stress-induced loss of body weight in marmosets in some facilities. The use of larger cages with additional environmental enrichment led to an increase in marmoset body weight while animals in small cages with less enrichment displayed stereotypic behaviors that were not observed in other groups [16]. It is essential that marmosets have access to thermoneutral materials to sit on and wood to scent mark on as this is a stress-relieving species-typical behavior. Pheromonal communication via scent marking is important for marmosets and other callitrichids, and thus an item that has been scent-marked (nest box and/or wood branch) should be transferred to the clean cage unit when cages are cleaned [6]. Research staff must always consider that routine husbandry procedures such as cage change and new pair formation may cause stress in marmosets that could impact experimental data [17]. Many facilities note relatively lower levels of stress in marmoset colonies if human caretakers are kept consistent, though the actual impact has yet to be studied. A common problem that occurs in marmoset colonies is the presence of a secretion-laden or “greasy” haircoat that can persist and occasionally lead to or exacerbate skin ulceration. Though this condition is seen frequently, no studies exist to date that have evaluated the incidence, etiology, or effective treatments. Increasing access to surfaces for scent marking and bathing marmosets with warm water can help restore haircoats to a more stable condition, though the effect may be temporary. Given the great variety in housing options and regulations for marmosets, it is clear that additional study is required in order to determine the optimal housing configuration for the species in biomedical research facilities. 

### 2.4. Handling and Acclimation

As a prey species, marmosets tend to be easily stressed by close contact with humans and are very quick to use flight to escape from any perceived predator, including human handlers. Flight, self-scratching, and scent marking are useful behavioral measures of stress for marmosets [18]. Training NHPs to cooperate with procedures that normally require hand capture via the use of positive reinforcement training (PRT) will significantly decrease the adverse effect of the procedures [19]. Routine preventative health procedures can be adapted and accomplished without handling animals. For example, marmosets can be trained to stand on a scale in the home cage to facilitate regular weight checks without the need for hand capture [13,20]. Positive reinforcement training can also be used to train marmosets to provide clinical samples, such as urine samples [18]. Specific recommendations for creating and implementing training programs for NHPs are described in Prescott 2005 [21]. Marmosets should be acclimated to all procedures that involve close interaction with a human handler or hand capture. Animals should be trained to regularly engage in routine veterinary procedures such as weight checks in the home cage to avoid the inherent stress of hand capture as much as possible. Minimizing the stress associated with hand capture will decrease the cumulative stress experienced by marmosets and may improve overall welfare. 

### 2.5. Diet

Marmoset nutritional requirements are not completely understood, though many past studies have attempted to define best practices for feeding marmosets. While the diet of wild marmosets is heavily based on exudates, fruits, and insects, the diets of captive marmosets need not exactly replicate this type of feeding [22]. A cafeteria-style diet with many supplemental diet items including fruit, vegetables, insects, or dairy products over a base of commercially prepared chow is often provided to marmoset families. This may result in individual animals having varying access to and intake of nutrients as the bowl is picked over by more dominant animals first before access is allowed to more subordinate animals. Captive marmosets vary greatly in their digestive efficiency and nutrient absorption [22]. The gut microbiome of captive marmosets has aspects of composition that are seen in human gastrointestinal disease [23]. Recommendations for marmoset diets are presented in *The Common Marmoset in Captivity and Biomedical Research*, Fitz et al., 2020, and Ross et al., 2020 [12,22,24,25]. Both obesity and a thin body condition are health issues that occur commonly within marmoset colonies and can adversely impact animal health and welfare. According to a survey of 39 institutions with 14 responses conducted by Ross et al. in 2020, average weights for adult male and female marmosets are 414.7 ± 2.9 g and 423.3 ± 7.9 g, respectively [25]. A different recent survey characterized nutrition programs, disease incidence, and other factors and found that obesity occurred more frequently in captive marmoset colonies at an average of 20% of the reporting institution’s marmosets compared to thin body condition at an average of 5% of animals [26]. It is imperative for each facility to assess the diet used by their marmoset source and adapt as needed for the colony on an ongoing basis using trends of disease, obesity, and other factors as indicators that dietary changes may be required. Recommendations and resources related to diet and metabolic management are presented in Table 2. 

## 3. Veterinary Care

As a species maintained in captivity, common marmosets are predisposed to a number of spontaneous diseases and other conditions that can negatively impact welfare in addition to survival and research use. A thorough overview of clinical illnesses that commonly impact marmosets is provided within the proceedings of the ILAR Roundtable Workshop “Care, Use and Welfare of Marmosets as Animal Models for Gene Editing-Based Biomedical Research” as well as within the individual publications of the 2020 marmoset special edition of the *ILAR Journal* [24,25,27]. The current publication will focus on recent updates to the marmoset veterinary care topics as well as the potential impacts that clinical issues can have on marmoset welfare and ways to mitigate these impacts. 

### 3.1. Gastrointestinal and Musculoskeletal Disease

Gastrointestinal (GI) diseases of varying scope and etiology that can impact survival and animal welfare have been documented to occur frequently in marmoset colonies worldwide [7,26,28,29]. The most devastating non-infectious presentation of GI disease is an inflammatory bowel disease (IBD)-like disease historically known as “marmoset wasting disease” that often presents with weight loss, muscle atrophy, diarrhea, anemia, hypoproteinemia, enteritis, among other signs, or with acute moribundity or death. This disease syndrome may present with GI signs only or, in some colonies, with concurrent musculoskeletal disease that may predispose affected animals to pain or pathologic fractures [30]. 

A definitive diagnosis of IBD-like disease can be difficult to attain as the gold standard for diagnosis has historically been the presence of chronic lymphoplasmacytic enterocolitis documented on postmortem examination of tissues. Recently, a technique for the collection of minimally invasive endoscopic biopsies of the upper and lower GI has been described for common marmosets [31]. It would be ideal to have a reliable diagnosis of this IBD-like syndrome based on clinical pathology and other noninvasive data. One retrospective study showed that low body weight (<325 g) and low serum albumin (<3.5 g/dL) could identify 100% of marmosets affected with concurrent bone and gastrointestinal disease [32]. The precise etiology of this IBD-like disease is unknown, though it is likely that it is multifactorial [12,24]. Treatment with glucocorticoids is one of the only therapies that has been systematically evaluated and shown to have a positive effect. Treatment of marmosets with IBD-like disease with 1 mg/kg prednisone followed by 0.5–0.75 mg total of budesonide orally significantly improved weight and serum albumin measures over time [33]. Tranexamic acid and supportive care measures may be helpful to treat IBD-like disease as well [34]. Commercial calprotectin fecal assays may be useful for identifying marmosets with IBD-like disease [35]. While it is clear that further study is required to establish evidence-based best practices for the diagnosis and management of IBD-like disease in marmosets, it is recommended that a trial of glucocorticoid therapy, ideally with budesonide from a reputable compounding pharmacy, be used to treat animals presenting with signs that fit the clinical picture described above. The author recommends proactive treatment with glucocorticoids as soon as IBD-like disease is suspected. 

Recently, novel GI disease presentations have been documented in marmoset colonies in Japan and the US that have resulted in relatively high morbidity and mortality due to complex duodenal dilation and duodenal stricture syndromes [36,37]. The duodenal dilation syndrome affecting 21.9% of necropsy cases at a facility in Japan was recently described. Clinical signs included vomiting, bloating, and weight loss presenting with significant dilation of the descending duodenum [36]. The duodenal ulceration/stricture syndrome was recently characterized in marmosets in a colony in the US using clinical signs including diarrhea, weight loss, or poor weight gain with concurrent serum chemistry and complete blood count abnormalities such as hypoalbuminemia, hypoglobulinemia, hypoproteinemia, hypocalcemia (total), elevated alkaline phosphatase, anemia, and occasionally leukocytosis [37]. The precise etiology of both syndromes has yet to be determined; however, an increased abundance of *Clostridium perfringens* present in duodenal ulcer/stricture cases is a possible root cause. Careful monitoring of trends in weight, bloodwork parameters such as albumin, hematocrit, hemoglobin, and calcium, and routine radiography can help identify marmosets with IBD-like and/or bone disease or duodenal disease early and treatments such as glucocorticoid therapy should be initiated as soon as possible to halt progression if possible. Any animal identified as potentially being affected should be monitored for pain and provided analgesia as needed. Ultimately, early recognition and management of these diseases is key to mitigating adverse health and welfare impacts.

### 3.2. Obesity

As referenced above in Section 2.3, marmoset health and welfare may be impacted by metabolic dysregulation resulting in obesity and potentially insulin resistance. Monitoring of body weight trends and clinical pathology data should begin early in life as marmosets with early-onset obesity show impaired glucose homeostasis by one year of age [38]. Additional cage space can be provided to encourage more caloric output via increased exercise. The pharmacokinetics of oral hypoglycemics such as acarbose and metformin have been analyzed and these medications may be compounded by a reliable pharmacy [39]. The use of calorically dense treats, especially those high in sugar such as marshmallows, should be avoided when at all possible as regular consumption of such food may result in a state of caloric excess. 

### 3.3. Reproduction and Genetic Management

Aspects of reproductive use can negatively impact the health and welfare of marmosets in breeding and transgenic colonies. Reliable contraception is critical to ensure that marmosets are not overproduced. Accurate calculations should be performed for each breeding colony to ensure that continued breeding is required. Each facility should have an active plan for genetic management to avoid in-breeding and potential founder effects. Reliable methods of contraception include vasectomy of the breeding male in the pair and careful use of cloprostenol given intramuscularly. Techniques for vasectomy of the male marmoset are outlined in Chapter 11 of *The Common Marmoset in Biomedical Research* [40]. Even when cloprostenol is administered appropriately and regularly, it is not successful at preventing 100% of pregnancies. More permanent methods of sterilization are recommended to control breeding in male–female pairs that are not intended to serve as breeding stock. If permanent methods that alter sex hormone levels, such as ovariectomy, are to be used to sterilize animals, the impact on potential future research use must be considered. 

Although common marmoset dams can give birth twice a year due to their ability to conceive in the weeks immediately following birth, short interbirth intervals are associated with higher infant mortality [41]. It may be beneficial to avoid breeding dams during postpartum ovulation to promote survival of future infants, though the optimal interbirth interval period has not been established. The author recommends that criteria be developed for continued use of marmoset females as breeders and for removal from breeding stock (Table 3). In addition to providing preventative health care and treatment of spontaneous disease and other health conditions, veterinary care teams must carefully monitor colony size (current and projected as applicable) and composition, and breeding pairs, and have an active plan for genetic management to avoid in-breeding and potential founder effects.

### 3.4. Pain Management and Surgical Support

The identification and management of pain, whether spontaneous, perioperative, or related to research manipulations, is key to optimizing the welfare of marmosets engaged in biomedical research. The development and validation of a pain scoring system such as that described for cynomolgus macaques in Paterson et al. would enhance the ability of marmoset users to adequately manage perioperative and spontaneous pain in marmosets [42]. The author encourages the use of a facility-specific pain assessment tool, especially for postoperative pain management, that can be updated as more evidence-based guidelines are published. (Table 4). Techniques for marmoset anesthesia, pain management, and surgical support are thoroughly described in Chapter 11 of *The Common Marmoset in Captivity and Biomedical Research*, Chapter 18 of *Anesthesia and Analgesia of Laboratory Animals*, and Goodroe et al., 2020 [40,43,44]. The author encourages all marmoset users to use the specific drugs and treatment regimens for the provision of anesthesia and analgesia described in these and other evidence-based guideline documents. Perioperative supportive care measures such as thermal support, species-specific tools for intubation and surgical manipulation, use of syringe pumps for intravenous fluid and drug administration, use of maropitant citrate to prevent nausea, and consistent monitoring and supportive care during the postoperative phase will optimize postoperative health and welfare outcomes. 

### 3.5. Quality of Life Monitoring 

The development of a clinical illness can impact marmoset welfare by decreasing overall quality of life (QoL) due to factors such as the development of discomfort or pain, decreased ability to perform normal behaviors, decreased or impaired social interactions, impaired psychological state, among others. Human health QoL measurement tools have been used in practice for many years and it has become more common to employ these tools in companion animal practice as well [45]. Quality of life committees formed for the monitoring of NHP QoL involve identifying stakeholders, including caretakers, veterinarians, researchers, pathologists, behaviorists, and others. Regular observations of animals along with scoring on a number of relevant parameters such as body weight loss, appetite, activity, medical conditions, interaction with cagemates, and many others facilitate discussions focused on overall QoL [46]. Formal quality of life committees have been employed successfully with colonies of research NHPs. Recommendations for quality of life parameters for NHPs have been established, as have humane endpoint guidelines [46,47]. The subsections above provide additional information regarding the common clinical issues that may impact marmoset welfare by decreasing overall QoL. 

## 4. Welfare Implications of Research Use: Select Models

A thorough review of the use of marmosets as animal models for biomedical and preclinical research is provided by Han et al. and *The Common Marmoset in Captivity and Biomedical Research,* among other publications [3,12]. While a comprehensive review of all potential research-related welfare impacts and refinements is outside of the scope of the current publication, suggested refinements for select common models will be presented below and are outlined in Table 5. Close communication among husbandry, veterinary, and research teams will result in the best animal welfare and study outcomes [48].

### 4.1. Sample Collection

Marmosets have long been used in preclinical research and maximum recommended blood volume and dosing volume limitations have been established [49]. A maximum blood volume of 3 mL is recommended for a single blood collection from an adult marmoset [50]. Techniques for collection of samples including blood samples, skin biopsies, and urine samples, among others have been published as well [10]. Many samples may be collected without sedation, which limits the impact of sedative drugs on sensitive parameters such as blood hormone levels. It is best practice for marmoset users to use the recommendations contained in these published guidelines when using new procedures at their respective institutions. Reference ranges for marmoset clinical pathology parameters have been published [51,52].

### 4.2. Aging

Due to their relatively short lifespan, marmosets have long been used as animal models of aging. This work has been well summarized previously in numerous publications [53,54]. The author recommends that accommodations be made to animal housing, handling, and routine health monitoring as marmosets age. For example, additional perches can be added to caging, along with soft couches or other soft surfaces to help alleviate any discomfort related to osteoarthritis that can be exacerbated by sitting on hard surfaces. The frequency of health monitoring should be increased depending on the specific condition of the animal and/or any abnormalities noted on physical exams or bloodwork. Both male and female marmosets may be reproductively active into their adult years, with dominant females observed to breed for up to 8 years in captivity [55]. This highlights the importance of reliable contraception for male–female pairs throughout adulthood.

### 4.3. Neuroscience

In recent decades, marmosets have emerged as critical nonhuman primate models for neuroscience research in a diverse array of fields, including visual processing, neurodegenerative disease (Parkinson’s), autoimmune-mediated inflammatory disorders (EAE/MS), language acquisition and development, social behavior, and many others [56,57,58]. Refinements to historical practices used in neuroscience research, such as water regulation, should be made for all marmosets engaged in this research in order to optimize welfare during use. For example, one recent study showed that the amount of water that each marmoset consumes varies greatly and therefore it is recommended to create individualized fluid regulation guidelines for every animal on water regulation protocols often used in neuroscience research [59]. Many studies in this area of research involve the use of advanced imaging techniques that historically may have involved sedating animals to facilitate imaging. Recently, techniques for acclimation of marmosets to a custom-made cradle and helmet for use in the MRI have allowed for specialized imaging of alert animals. Positive reinforcement training has also been used in the form of high-value treats [60]. Chair training marmosets has historically involved the use of hand capture to move the marmoset from the transport box to the chair. A method of using a custom squeeze-walled transfer box to move the marmoset into the chair without the need for hand capture has recently been described [61]. Videorecording and novel methods of analysis have also been applied to home-cage monitoring of animal activity to potentially eliminate the need for handling animals for some neuroscience studies [62,63]. Ultimately, the author strongly encourages refinements to research-related procedures for neuroscience that minimize the use of hand capture and rely on acclimation and positive reinforcement training to encourage animal engagement and to minimize adverse welfare impacts. 

### 4.4. Transgenics

Assisted reproductive techniques and technologies have advanced to allow for the reliable generation of transgenic NHP models and an associated body of work describing techniques successful in the common marmoset has been published in recent years [64,65,66]. The use of marmosets to create transgenic models requires unique ethical consideration and oversight. Transgenic marmoset models, including those of neurodegenerative disease among others, have been created by research teams around the globe [67]. This new age raises ethical concerns related to the use of NHPs to create transgenic models outlined in Prescott, 2020 [68,69]. While these technologies have been successful, they remain inefficient, which inherently results in the usage of a relatively higher number of NHPs compared to other areas of research. Interestingly, there appears to be individual variation in how well embryo donors are able to tolerate repeated blood sampling and anesthetic events required for assisted reproductive procedures [70]. Given the relatively high number of animals required to maintain a transgenic marmoset colony, any steps to refine the process will have a great impact on improving overall welfare. 

## 5. Conclusions and Future Directions

While further study is undoubtedly needed in many areas of husbandry, veterinary care, and research use to optimize marmoset health and welfare, some data have been produced in recent decades to allow for initial evidence-based guidelines on these topics to emerge. A framework for these guidelines and resources is highlighted in Table 1, Table 2 and Table 3. It is incumbent upon all marmoset users, including husbandry teams, veterinarians, researchers, and ethical review body members, to consider the aspects discussed above when designing and implementing a marmoset research program. Close and ongoing communication among husbandry, veterinary, and research teams will result in the best animal welfare and study outcomes. With refinements in husbandry practices, preventative health care, and research techniques, welfare impacts to marmosets will be mitigated to the fullest extent possible and the data generated via their research use will be improved. This need for further study and guidelines is especially dire for common marmoset health conditions such as IBD-like disease as this syndrome has profound health and welfare impacts on marmoset colonies worldwide. The author encourages all marmoset users to collaborate with the global marmoset community to work towards a complete understanding of this syndrome (and others) that adversely impact marmoset survival, research use, and welfare. Specific guideline documents should be created for each program that outline evidence-based best practices for marmosets and tailor each to the specific program’s constraints and needs.

## Figures and Tables

**Table 1 vetsci-10-00568-t001:** Recommended steps to optimize marmoset health and welfare during colony preparation and maintenance.

Timeline	Action
Prior to colony formation	Consult with existing marmoset colony management teams on facility layout and cage environment plans and training program development Conduct animal vendor assessment Determine facility layout and cage environment Provide opportunities for husbandry and veterinary staff training, ideally involving shadowing at an existing marmoset facility Develop evidence-based researcher training sessions
During colony formation	• Verify transport conditions with vendor and transporter • Create and train staff on specific arrival and acclimation plan for new animals • Keep initial diet as consistent with source facility as possible and transition the diet slowly • Quarantine ○ Create plan for specific facility based on published guidelines ○ Use a formal monitoring system with a minimum of two daily observations to ensure timely interventions as needed
Colony maintenance	Monitor impact of facility-specific diet Analyze weight trends and colony clinical pathology data Implement a validated reproductive management plan Use acclimation and positive reinforcement training as part of all hand capture events Monitor caging, animal handling, and social groups to prevent injuries

**Table 2 vetsci-10-00568-t002:** Facility, housing, and husbandry recommendations.

Topic	Recommendations
Housing	Use caging from a vendor designated as marmoset-specific OR work with vendor to create custom caging (avoid repurposing caging originally manufactured for other species) Monitor weight and health trends within rooms and colony if changes are made to macro- or microenvironment
Diet	Assess diet from source/vendor facility, recreate if possible Routinely evaluate food intake Monitor weight trends within colony to ensure individual animals are not trending towards either obesity or low body weight; reference survey averages for adult weights Use human-grade produce and clean with a validated method prior to feeding Ensure that produce is appropriately cleaned and managed Remove any produce or moist chow from cage before mold develops Avoid regular use of high-calorie treats
Handling	Use of acclimation prior to handling for procedures Use of positive reinforcement training for all procedures Minimize hand capture events

**Table 3 vetsci-10-00568-t003:** Veterinary care recommendations.

Condition/Topic	Recommendations
Gastrointestinal and musculoskeletal disease	Ensure that diet is based on published recommendations Monitor weight trends Use glucocorticoid (recommend budesonide compounded from a reliable pharmacy) to treat as needed for IBD-like disease Survey radiography to evaluate for bone disease Provide oral vitamin D and calcium supplementation as needed (specific dosage levels outlined in Fitz et al., 2020) [24] Additional diagnostics for any suspected duodenal disease: Endoscopic biopsy of GI Radiography (barium as needed) Ultrasonography
Infectious disease	Vendor assessment Quarantine period based on published recommendations Routine health monitoring Train staff to avoid contact with marmosets if they (staff) are sick
Reproductive	Create formal colony and genetic management plan Use effective contraception to control breeding as needed (cloprostenol, vasectomy) Develop criteria for removal of females from breeding stock Prolong interbirth interval where possible
Metabolic	Monitor body weight and body condition, identify intervention point for colony Monitor clinical pathology data and HgbA_1c_ Increase cage space to allow for increased caloric output Treatment with oral hypoglycemic agents
Pain management and surgical support	Develop and validate pain scoring system specific to colony and research use Use specific drugs and dosing regimens recommended in evidence-based guidelines to provide effective anesthesia and analgesia to marmosets

**Table 4 vetsci-10-00568-t004:** Other recommendations.

Topic	Recommendations
Specific welfare indicators	Access to social, physical, and food enrichment Negative: flight, self-scratching, and scent marking
Quality of life management	Develop a program-specific quality of life assessment and monitoring tool Encourage staff engagement in QoL monitoring and develop a culture of care program

**Table 5 vetsci-10-00568-t005:** Refinements to marmoset research models.

Intervention/Condition	Recommendations
Aging	Modify housing as needed Modify handling practices Increase frequency of preventative health monitoring
Neuroscience	Customize water regulation paradigms based on individual consumption patterns Use alert advanced imaging facilitated by acclimation and positive reinforcement training Use home cage videorecording Minimize hand capture
Transgenic	Provide education on NHP transgenic models to IACUC/ethical review body Require phenotyping of new transgenic models Provide supportive care and additional monitoring as needed for transgenic animals

## Data Availability

No new data were created or analyzed in this study. Data sharing is not applicable to this article.

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
