# Peer review of "Review of Environmental and Health Factors Impacting Captive Common Marmoset Welfare in the Biomedical Research Setting"

_vetsci, 2023, doi:10.3390/vetsci10090568_

Round 1
Reviewer 1 Report
This paper provides a review of some of the environmental and health factors that may impact welfare of captive common marmosets in biomedical research facilities. Overall, the author does a nice job summarizing available information in the literature and presenting tables to use as guidelines in determining appropriate care. Appropriate references are provided to guide the reader to more detailed and primary source information as needed. Major factors that could impact welfare are covered and the main research uses are described.
Specific comments/suggestions:
· Important environmental factors such: the number of animals in a cage, type of social housing, number of animals and cages in a room, environmental noise, proximity to other species, are not discussed. While a detailed discussion is outside the scope of this paper, at a minimum when discussing housing, the importance of these factors should be mentioned.
· Facility, housing and husbandry and Table 1. It would be ideal if people were encouraged to consult with existing marmoset colonies particularly prior to colony formation on factors such as facility layout and cage environment and the development of training.
· 2.2 Animal transport and quarantine. There is no mention of appropriate housing (single, paired, social) during transport and quarantine.
· Line 129-130: In this case, increased body weight is presumably a positive outcome. As obesity can be an issue with captive marmosets, please clarify this conclusion.
· Line 129-131: In addition to the cages being smaller, according to the referenced publication, there was less environmental enrichment in the smaller cages and no nest boxes. These facts should be made clear.
· Table 2, housing first bullet. In parenthesis I assume this should say: avoid repurposing caging originally manufactured for other species.
· Table 2, diet, second bullet. Should monitor for underweight as well as obesity. Though perhaps less common it is an important indicator of health.
· Table 2, diet, second bullet. Perhaps recommend that food intake be evaluated routinely.
· 3.2 Obesity and Table 2, diet and/or in related text. Please add a sentence imploring people to refrain from, or at least limit commonly used sugary treats such as marshmallows.
· 3.3 Reproduction and genetic management. At the end of the first paragraph more permanent methods of sterilization are recommended to control breeding in pairs that are not intended to serve as breeding stock. Please include a word of caution here that some such methods may potentially interfere with research use of these animals. For example, ovariectomized animals may not be appropriate for all types of research.
· Table 3, GI and musculoskeletal disease, 4th bullet. Survey radiography for older animals is suggested to evaluate for bone disease however in the text, the bone disease that is described is not to my knowledge age-related. Why is the recommendation specifically for older animals? And if this will remain in the table, what is the definition of older?
· Table 3, GI and musculoskeletal disease, 5th bullet. Are there recommendations for when these supplements are needed and what doses/forms are appropriate.
· Table 3. Why isn’t pain management included in this table?
· 4.3 Neuroscience. The following sentence: “Close communication among 363 husbandry, veterinary, and research teams will result in the best animal welfare and study 364 outcomes.” Towards the end of this section is extremely important for all areas of research, not just neuroscience. I suggest moving this sentence (possibly to the paragraph immediately below 4. Welfare implications of research use: select models), and also restating it in the conclusion paragraph.
· 4.4 Transgenics. The last sentence in this paragraph seems to be missing something.
Author Response
Specific comments/suggestions:
- Important environmental factors such: the number of animals in a cage, type of social housing, number of animals and cages in a room, environmental noise, proximity to other species, are not discussed. While a detailed discussion is outside the scope of this paper, at a minimum when discussing housing, the importance of these factors should be mentioned.
A sentence mentioning these factors has been added to section 2.3.
- Facility, housing and husbandry and Table 1. It would be ideal if people were encouraged to consult with existing marmoset colonies particularly prior to colony formation on factors such as facility layout and cage environment and the development of training.
This recommendation has been added to Table 1.
- 2.2 Animal transport and quarantine. There is no mention of appropriate housing (single, paired, social) during transport and quarantine.
This has been addressed generally in section 2.2 with a reference to local, national, and international guidelines. I have only received/shipped animals in individual transport containers (excluding infants & dams) but am happy to incorporate other recommendations on paired/social shipment if it is done routinely elsewhere.
- Line 129-130: In this case, increased body weight is presumably a positive outcome. As obesity can be an issue with captive marmosets, please clarify this conclusion.
This has been clarified.
- Line 129-131: In addition to the cages being smaller, according to the referenced publication, there was less environmental enrichment in the smaller cages and no nest boxes. These facts should be made clear.
The word “additional” was added to the sentence to highlight that cage enrichment, in addition to cage size, was enhanced for the larger cages.
- Table 2, housing first bullet. In parenthesis I assume this should say: avoid repurposing caging originally manufactured for other species.
This has been corrected.
- Table 2, diet, second bullet. Should monitor for underweight as well as obesity. Though perhaps less common it is an important indicator of health.
A statement has been added in parenthesis to clarify that both low and high body weight trends should be flagged.
- Table 2, diet, second bullet. Perhaps recommend that food intake be evaluated routinely.
This has been added to Table 2.
- 3.2 Obesity and Table 2, diet and/or in related text. Please add a sentence imploring people to refrain from, or at least limit commonly used sugary treats such as marshmallows.
Comments related to this point have been added to both section 3.2 and Table 2.
- 3.3 Reproduction and genetic management. At the end of the first paragraph more permanent methods of sterilization are recommended to control breeding in pairs that are not intended to serve as breeding stock. Please include a word of caution here that some such methods may potentially interfere with research use of these animals. For example, ovariectomized animals may not be appropriate for all types of research.
This has been addressed in section 3.3.
- Table 3, GI and musculoskeletal disease, 4thbullet. Survey radiography for older animals is suggested to evaluate for bone disease however in the text, the bone disease that is described is not to my knowledge age-related. Why is the recommendation specifically for older animals? And if this will remain in the table, what is the definition of older?
The limitation of this recommendation to older animals has been removed. I would recommend this for at-risk animals vs. older animals. Each colony should define what “at-risk” means based on their clin path, necropsy, and colony health data/trends.
- Table 3, GI and musculoskeletal disease, 5thbullet. Are there recommendations for when these supplements are needed and what doses/forms are appropriate.
Yes – this is described in Fitz et al 2020. This reference has been added to the table.
- Table 3. Why isn’t pain management included in this table?
It has been included.
- 4.3 Neuroscience. The following sentence: “Close communication among 363 husbandry, veterinary, and research teams will result in the best animal welfare and study 364 outcomes.” Towards the end of this section is extremely important for all areas of research, not just neuroscience. I suggest moving this sentence (possibly to the paragraph immediately below 4. Welfare implications of research use: select models), and also restating it in the conclusion paragraph.
This has been removed from the neuroscience section and has been added to the intro paragraph of section 4 as well as the conclusion paragraph.
- 4.4 Transgenics. The last sentence in this paragraph seems to be missing something.
The remainder of the sentence has been added.

Reviewer 2 Report
This is a nice review, clear and well-written.
3.4 Pain management.
Some species have unique pain management issues. What drugs are safe for pain management in marmosets? Are there any references.
This could be expanded to include surgical needs for this species. Are there any unique surgical concerns (thermal regulation, intubation etc).
Section 4. could be expanded a little.
4.1 Please discuss sample volumes (quantity of blood or other tissues that can be reasonably collected; can blood or other samples be collected without sedation which could affect ovarian cyclicity;
4.2 Aging studies are important because they can be extrapolated to human health. This could be expanded. What is known about reproductive aging in female and male marmosets? Do they go through marmoset menopause (marmopause?)
There is currently a shortage of nonhuman primates due to the use for infectious diesase. Can the marmoset be used as an alternative for other species for covid or other studies?
Author Response
3.4 Pain management.
Some species have unique pain management issues. What drugs are safe for pain management in marmosets? Are there any references.
This topic is covered extensively in other texts including: The Common Marmoset in Captivity and Biomedical Research (Chapter 11), Anesthesia and Analgesia of Laboratory Animals (Chapter 18), and Goodroe et al 2020. References to these publications have been added to the text as has a recommendation that marmoset users base their specific anesthesia and analgesia practices on these and other evidence-based guidelines. It is outside of the scope of this publication to describe/summarize the specific recommendations that are covered in these other texts.
This could be expanded to include surgical needs for this species. Are there any unique surgical concerns (thermal regulation, intubation etc).
The section has been expanded a bit to reference other texts that address surgical needs. Specific perioperative concerns have been highlighted in this text as well.
Section 4. could be expanded a little.
I agree, however I think that the topic could easily be expanded to a full review manuscript in itself. With this section, my intention was to bring up the topics most likely to impact welfare in some of the most common areas of research and to point the reader to the many reference that already cover the topics highlighted in further detail.
4.1 Please discuss sample volumes (quantity of blood or other tissues that can be reasonably collected; can blood or other samples be collected without sedation which could affect ovarian cyclicity;
A reference to a maximum single blood draw volume of 3 ml has been added to this section as has a reference to the fact that some samples may be collected without sedation.
4.2 Aging studies are important because they can be extrapolated to human health. This could be expanded. What is known about reproductive aging in female and male marmosets? Do they go through marmoset menopause (marmopause?)
Female marmosets do undergo reproductive senescence later in life (past the point of being considered “aged”). I have added a comment highlighting this into this section.
There is currently a shortage of nonhuman primates due to the use for infectious diesase. Can the marmoset be used as an alternative for other species for covid or other studies?
Marmosets seem to be a poor substitute for macaques as a COVID model. Marmosets demonstrate mild infection after SARS-CoV-2 infection whereas macaques and baboons develop moderate and severe infection/pathology respectively. (Singh DK et al. Responses to acute infection with SARS-CoV-2 in the lungs of rhesus macaques, baboons, and marmosets. Nature Microbiology. 2021)
